# Computational and ADMET Predictions of Novel Compounds as Dual Inhibitors of BuChE and GSK-3β to Combat Alzheimer’s Disease

**DOI:** 10.3390/pharmaceutics16080991

**Published:** 2024-07-26

**Authors:** Saurabh G. Londhe, Vinayak Walhekar, Mangala Shenoy, Suvarna G. Kini, Marcus T. Scotti, Luciana Scotti, Dileep Kumar

**Affiliations:** 1Department of Pharmaceutical Chemistry, BVDU’s Poona College of Pharmacy, Pune 411038, Indiavinayakwalhekar64@gmail.com (V.W.); 2Department of Pharmaceutical Chemistry, Manipal College of Pharmaceutical Sciences, Manipal Academy of Higher Education, Manipal 576104, India; 3Health Sci. Center, Federal University of Paraíba, João Pessoa 50670-910, PB, Brazil; mtscotti@gmail.com (M.T.S.); luciana.scotti@gmail.com (L.S.); 4Teaching and Research Management—University Hospital, Federal University of Paraíba, Campus I, João Pessoa 58051-900, PB, Brazil; 5Department of Entomology, University of California, Davis, One Shields Ave, Davis, CA 95616, USA; 6UC Davis Comprehensive Cancer Centre, University of California, Davis, One Shields Ave, Davis, CA 95616, USA

**Keywords:** Alzheimer’s disease, molecular docking, molecular dynamics, GSK-3β, BuChE, tacrine

## Abstract

Background: Alzheimer’s disease is a serious and widespread neurodegenerative illness in the modern healthcare scenario. GSK-3β and BuChE are prominent enzymatic targets associated with Alzheimer’s disease. Co-targeting GSK3β and BChE in Alzheimer’s disease helps to modify disease progression and enhance cognitive function by addressing both tau pathology and cholinergic deficits. However, the treatment arsenal for Alzheimer’s disease is extremely inadequate, with present medications displaying dismal success in treating this never-ending ailment. To create novel dual inhibitors, we have used molecular docking and dynamics analysis. Our focus was on analogs formed from the fusion of tacrine and amantadine ureido, specifically tailored to target GSK-3β and BuChE. Methods: In the following study, molecular docking was executed by employing AutoDock Vina and molecular dynamics and ADMET predictions were performed using the Desmond and Qikprop modules of Schrödinger. Results: Our findings unveiled that compounds DKS1 and DKS4 exhibited extraordinary molecular interactions within the active domains of GSK-3β and BuChE, respectively. These compounds engaged in highly favorable interactions with critical amino acids, including Lys85, Val135, Asp133, and Asp200, and His438, Ser198, and Thr120, yielding encouraging docking energies of −9.6 and −12.3 kcal/mol. Additionally, through extensive molecular dynamics simulations spanning a 100 ns trajectory, we established the robust stability of ligands DKS1 and DKS4 within the active pockets of GSK-3β and AChE. Particularly noteworthy was DKS5, which exhibited an outstanding human oral absorption rate of 79.792%, transcending the absorption rates observed for other molecules in our study. Conclusion: In summary, our in silico findings have illuminated the potential of our meticulously designed molecules as groundbreaking agents in the fight against Alzheimer’s disease, capable of simultaneously inhibiting both GSK-3β and BuChE.

## 1. Introduction

Among the multitude of neurodegenerative conditions, AD stands out as an inexorable and progressive disorder of the CNS. It is characterized by the gradual deterioration of neurons and is associated with advanced age [1]. Notably, aberrant protein processes play a significant role, where β-amyloid and abnormally hyperphosphorylated tau proteins disrupt the normal regulation and accumulate within the brains of affected individuals. These aggregates manifest as amyloid senile plaques and discrete neurofibrillary tangles [2].

GSK-3β is a serine/threonine kinase with multifaceted roles, primarily localized within the CNS. It has been extensively investigated and recognized for its pivotal role in regulating tau phosphorylation, a process crucial to neurodegenerative pathology [3]. Stated causes foster the formation of NFTs [4]. It is noteworthy that these events transpire not only in physiological contexts but also in pathological conditions. Substantial research endeavors encompassing preclinical assessments and early-phase clinical trials have consistently underscored GSK-3β’s potential as a compelling therapeutic target for AD. These investigations have substantiated GSK-3β’s role in AD pathology and its viability as an intervention target. The outcomes of these studies collectively suggest that directing therapeutic efforts toward GSK-3β holds considerable promise for devising innovative treatments aimed at retarding the advancement of AD. Recent scientific investigations have unveiled compelling evidence supporting the utility of GSK-3β inhibitors as a plausible avenue to restore the equilibrium from neurodegeneration to neurogenesis. Contemporary findings have showcased the capacity of GSK-3βIs to modulate this delicate equilibrium in both controlled in vitro laboratory settings and in vivo living organisms. These inquiries impart potential insights into the therapeutic prospects of GSK-3β inhibition, delineating its potential to facilitate neurogenesis while counteracting neurodegenerative cascades [5].

BuChE plays a significant role in AD as it is implicated in the degradation of acetylcholine, a neurotransmitter crucial for cognitive functions. In the brains of individuals with AD, there is a reduction in the levels of acetylcholine due to the loss of cholinergic neurons and decreased release of acetylcholine. BuChE, which is predominantly found in the brain, is one of the enzymes responsible for breaking down acetylcholine. The decrease in acetylcholine levels contributes to the cognitive deficits observed in AD, including memory impairment and declining mental function [6]. This enzyme’s action accelerates the breakdown of acetylcholine, which is already diminished in AD, leading to an even more pronounced neurotransmitter deficit. Research has shown that the increased activity of BuChE in the brains of individuals with AD is associated with the progression of the disease and the severity of cognitive impairment. Therefore, BuChE inhibitors have been investigated as potential therapeutic targets for AD. By inhibiting BuChE’s activity, it is possible to slow down the breakdown of acetylcholine, thus increasing its availability in the brain and potentially ameliorating some of the cognitive symptoms associated with AD [7].

BuChE affects neurotransmitter breakdown, while GSK-3β influences critical signaling pathways within neurons. Researchers are exploring interventions targeting both BuChE and GSK-3β as potential therapeutic strategies for AD. For instance, drugs targeting BuChE inhibition or GSK-3β modulation might be developed to mitigate the cognitive and pathological aspects of the disease.

Currently, no clinically available drug blocks both BuChE and GSK-3β simultaneously. Given the severity of AD, ongoing research is dedicated to formulating new compounds capable of dual inhibition, targeting both BuChE and GSK-3β. This endeavor is pursued through a structure-based drug design methodology, with the goal of creating potential anti-Alzheimer agents. ADMET studies are integral to in silico drug discovery, as they enable the forecasting of drug candidate pharmacokinetic and safety profiles, thereby mitigating the risk of late-stage attrition. Additionally, these compounds will undergo rigorous in silico ADMET profiling to meticulously assess their pharmacokinetic and safety profiles. The ultimate objective is to identify a lead molecule from these efforts, which can then undergo further development into a promising drug candidate.

Two frequently employed tactics for the development of multitarget drugs encompass the utilization of merged and hybrid ligands. In our investigation, our emphasis was directed towards the amalgamation of tacrine and adamantane, giving rise to a hybrid ligand. This strategic approach involves the construction of substantial molecules employing linkers, thereby conferring upon a singular molecule the ability to manifest two discrete pharmacological activities. The hybrid ligand methodology extends a distinctive prospect for the synthesis of compounds endowed with heightened therapeutic potential. By integrating two pharmacologically active constituents, our objective rested upon harnessing the synergistic effects of both, thereby expanding the therapeutic reach of the resultant molecule (Figure 1).

To substantiate our hypothesis in the current research, we strategically employed scaffold hopping and molecular hybridization methodologies for the design of novel molecules. These molecules were meticulously crafted as analogs (uredio, amide and schiff base), interlinking pyridopyrimidine and thiazole moieties via uredio and amide linkage with a central phenyl ring as a spacer. The unique molecular design is based on the fact that 2-[2-(cyclopropylcarbonylamino)pyridin-4-yl]-4-methoxy-1,3-thiazole-5-carboxamide and tacrine are the FDA-authorized drugs. However, our approach involved their modification to kindle a new class of compounds with the potential for dual selectivity towards GSK-3β and BuChE. The structures of the designed molecules are portrayed in Figure 2.

## 2. Materials and Methods

### 2.1. Molecular Docking

Molecular docking analysis was ascertained to obtain knowledge of the ligands’ interactions with active site amino acids, as well as the resulting conformations that contribute to the stability of the docked complex. For this purpose, we employed the AutoDock Vina 1.2.0 module from The Scripps Research Institute package, a well-established tool for molecular docking studies [8]. The X-ray crystal structures used were sourced from the RCSB Protein Data Bank, specifically PDB IDs 4PTC (GSK-3β) and 4BDS (BuChE), featuring resolutions of 2.71 and 2.10 Å, respectively. The structures were drawn using ChemBioDraw Ultra 14.0. Subsequent ligand preparation and energy minimization were executed with ChemBio3D Ultra 14.0. On the other hand, protein preparation was performed using AutoDockTools-1.5.6. Validation of the docking protocol was undertaken by the redocking of the co-crystal ligands where the inhibitors were extracted from the proteins and further converted to .pdbqt, and docking was executed by the application of the vina command. The metrics for the validation were the RMSD of the redocked ligand and that the co-crystal ligand should be less than 1 Å. The efficiency of compound interactions with the macromolecule was quantified in kcal/mol, as recorded in the log file generated during the simulation studies. For the visualization of the docking outcomes, we employed PyMOL 2.1.1 and Discovery Studio Visualizer 2020 [9,10].

### 2.2. Molecular Dynamics Simulation

We performed a molecular dynamics (MD) simulation study on the promising compounds identified through the docking analysis, employing the Desmond tool within the Schrödinger software’s Maestro 11.8 platform. This simulation was carried out over a duration of 100 nanoseconds (ns). The study encompassed three main phases: Firstly, the system was constructed using the system builder module. Subsequently, a minimization step was executed to optimize the molecular arrangement and eliminate unfavorable contacts. For the MD simulation itself, a three-step process was followed. System Setup: The XP docked complexes were employed as the starting point. The simulation environment was established using a predefined SPC solvent model. The system was designed under orthorhombic boundary conditions, creating an appropriate model for the simulation. Energy Minimization: Prior to commencing the MD simulation, an energy minimization procedure was performed. This step aimed to reduce potential energy conflicts within the system, ensuring a stable starting point for the simulation. MD Simulation: The simulation study was carried out using the NPT ensemble class, maintaining a temperature of 300 K and a pressure of 1 bar. This ensemble closely resembles real-world conditions, allowing the molecules to interact with the environment in a dynamic manner. In summary, our investigation encompassed the construction of the simulation system, energy minimization, and the actual MD simulation, which was conducted using the Desmond tool within the Maestro 11.8 Schrödinger software [11].

### 2.3. Molecular Dynamic Cross-Correlation and Principal Component (PCA) Scrutiny

Principal component analysis is a mathematical algorithm wherein the data are reduced to numerical values retaining the variations of the data. A 2D or 3D graph can be plotted using the data obtained. It is basically a method of reducing the dimensionality of a data set by converting a large set of variables into a small set without altering the information in the large set. In this study, we have used all the atoms obtained from MD analysis for the PCA analysis. This study was performed using the Schrodinger software 11.8 [12,13].

### 2.4. In Silico ADMET Profile

The molecules designed in this study underwent screening for their in silico ADMET properties using the QikProp module from the Schrodinger package [14]. Fourteen ligands with 3D structures were employed for docking, enabling the prediction of various essential physical, chemical, and pharmacokinetic parameters. The primary objectives of the study encompassed the forecasting of several critical factors, including adherence to Lipinski’s rule of five, polar surface area, count of rotatable bonds, permeability through the central nervous system, partition coefficient between the brain and blood, aqueous solubility, apparent MDCK cell permeability, binding to human serum albumin, permeability through Caco-2 cells, and the percentage of human oral absorption. These parameters hold significant importance in the evaluation of drug-like characteristics and the pharmacokinetic profiles of potential compounds under consideration.

## 3. Results

### 3.1. Molecular Docking

#### 3.1.1. X-ray Crystal and Active Site Topology of GSK-3β and AChE

Protein GSK-3β belongs to the category of serine threonine kinases. Its X-Ray crystal structure (PDB ID: 4PTC) was utilized for the purpose of molecular docking. The crystal structure of GSK-3β comprises four essential segments: the N-terminal, C-terminal, hinge region, and activation loop [15]. The upper segment of the structure consists of antiparallel β strands, forming a small component of the protein. This region, known as the N-terminal (Lys36-Leu132), is stabilized by intermolecular hydrogen bonding. The larger part of the structure is composed of a spiral α-helix referred to as the C-terminal (Val139-His831) [16]. Connecting these two terminals is the hinge region (Asp133-Thr138), which extends further into the flexible region known as the activation loop. The activation loop is comprised of two critical motifs: DFG (Asp200, Phe201, and Gly202) and HRD (His179, Arg180, and Asp181). These motifs play a pivotal role in determining the specificity of inhibitors [17]. The region between the N-terminal and C-terminal constitutes the active site. This is where ATP binds, involving the N- and C-terminals, hinge region, and the DFG motif, as depicted in Figure 3.

The protein BuChE belongs to the cholinesterase family and was utilized in molecular docking simulations based on its X-ray crystal structure (PDB ID: 4BDS). Notably, BuChE possesses an ellipsoidal conformation with distinct topological features. This protein is categorized as α/β protein, featuring a central mixed β-sheet consisting of twelve strands, surrounded by fourteen α-helices. The most remarkable architectural characteristic of BuChE is its deep and narrow gorge, serving as a conduit to the active site. This intriguing attribute is defined by a lining formed by fourteen conserved aromatic residues, arranged in a ring-like pattern. The BuChE homodimer structure, in which two subunits are symmetrically related through a crystallographic two-fold axis, is stabilized by a four-helix bundle. This bundle comprises helices αF′3 and αH from each subunit, and it is responsible for the interconnection between the subunits [18]. In essence, the distinctive structural features of BuChE, including its ellipsoidal shape, the arrangement of α-helices and β-strands, the presence of a narrow gorge with conserved aromatic residues, and the homodimeric assembly, collectively contribute to its functional properties and interaction capabilities (Figure 4).

In the context of the study, the ligand 2WE, co-crystallized with GSK-3β, established three distinct polar hydrogen bond interactions with key amino acids: Lys85, Val135, and Asp200. These interactions occurred within the catalytic pore, hinge region, and DFG motif of the protein, following a type-one binding pattern. Similarly, the ligand THA, co-crystallized with BuChE, engaged in hydrogen bond interactions with His438, a vital residue located within the protein’s active site.

To validate the reliability of the docking protocol employed, a re-docking experiment was conducted. The co-crystallized ligands, namely **2WE (associated with GSK-3β) and THA (Tacrine, associated with BuChE)**, were docked back into their respective protein structures (Figure 5). The outcomes of this re-docking process revealed the molecular interactions between the compounds and the amino acids within the active sites of the proteins. These interactions were accompanied by corresponding docking energy scores, which are detailed in Table 1 of the study. This procedure served as a means of authenticating the efficacy of the docking approach and the accuracy of the obtained binding interactions. All the interactions, namely the polar hydrogen bond and the hydrophobic (π-π, π-cation and π-alkyl) and electrostatic contacts generated during the process of molecular docking, have importance, where hydrogen bond interactions are vital for strong stability, hydrophobic contacts maintain the positioning of the molecule in the binding cavity, and electrostatic contacts are the charge interactions requires for anchoring the heterocyclic ring motifs of the ligand discretely.

#### 3.1.2. GSK-3β Kinase

Molecule DST-1, with a docking energy of −9.6 kcal.mol^−1^, appropriately occupied the active site of GSK-3β kinase. The nitrogen of the pyridopyrimidine moiety engrossed the hinge region by developing two hydrogen bond interactions with the carbonyl oxygen of Val135 and Asp133; hydrogen and oxygen of the amide functionality established three polar hydrogen bond contacts with oxygen of the carboxylic part of Asp200 in the DFG motif and Asn186. Moreover, the nitrogen of the thiazole was engaged with hydrogen bond interaction with Gln185 in the side chain. The pyridopyrimidine ring established π-sigma interactions with Leu188 and the central phenyl ring generated π-π staking contacts with Phe67 in the hydrophobic pocket of GSK-3β, as depicted in Figure 6a.

Compounds DKS-5 and -6 tenanted the active site of the protein, portraying similar pattern of interactions with corresponding docking energies of −9.1 and −8.6 kcal.mol^−1^. The oxygen of the amide carbonyl was engaged with hydrogen bond interactions with Val135 in the hinge region. The 3rd and 8th nitrogen of the pyridopyrimidine occupied the DFG and HRD motifs by establishing three polar hydrogen bond interactions with Asp200 and Gln185; NH of the uredio functionality also generated hydrogen bond interaction with Asn186. Furthermore, the central phenyl and thiazole rings were surrounded by the aliphatic amino acids, establishing π-alkyl contacts with Ile62, Val70, Ala83, Lys85, Cys199, and Arg141, as depicted in Figure 6a–c.

Molecule-entitled DKS-8 well settled in the active site of the kinase; the amide carbonyl established two hydrogen bond interactions in the hinge region with Asp133 and Val135. The urea, 3rd, and 8th nitrogen of the pyridopyrimidine ring developed three hydrogen bond interactions with Asp200, Asn186, and Gln185 in the DFG, HRD, and side chain regions with a docking energy of −9.6 kcal.mol^−1^, as shown in Figure 6d.

#### 3.1.3. BuChE

Compounds DKS-2 with a docking energy of −11.4 kcal.mol^−1^ resided in the active cavity of BuChE. The 3rd and 8th nitrogen of the pyridopyrimidine generated two hydrogen bond interactions with His438 and Tyr332. Carbonyl oxygen of the urea functionality established two hydrogen bond interactions with Ser198 and His438; moreover, the pyridopyrimidine and phenyl rings developed π-cation interactions with Trp82, Gly115, and Ala328, and the thiazole ring also developed π-cation contacts with Gly116 in the hydrophobic cavity of the protein, as depicted in Figure 7a.

Molecule DKS-4 occupied the active groove of the protein, with a corresponding docking energy of −12.3 kcal.mol^−1^. The carbonyl of the second urideo functionality established three hydrogen bond contacts with Ser198 and His438; amide functionality developed polar hydrogen bond interaction with Tyr332. Additionally, -CF3, Cl, and the phenyl functionalities occupied the hydrophobic cavity, generating π-cation and π-π staking interactions with Gly116, Ser287, and Phe329; the sulfur of the thiazole established π-sulfur interactions with Trp82, as stated in Figure 7b.

Molecules DKS-5 and -11, with corresponding docking energies of −12.1 and −11.7 kcal.mol^−1^, resided in the active site of the protein by producing numerous interactions. The -NH of the amide generated hydrogen bond contacts with His438, and the two nitrogens of the urea group established three hydrogen bond interactions with Asp70 and Tyr332. Moreover, the phenyl, central phenyl, and the thiazole rings developed π-π staking and π-alkyl hydrophobic interactions with Trp231, Phe329, Gly439, Trp82, Ala328, and Leu286. Similarly, both amide functional groups of the 11 engendered polar hydrogen bond contacts with Ser198, His438, and Ala328. Furthermore, the pyridopyrimidine, thiazole, and trifluoromethyl phenyl groups developed π-π staking and π-alkyl hydrophobic contacts with Gly116, Trp82, and Ala277, as shown in Figure 7c,d.

### 3.2. Molecular Dynamics Analysis

MD represents an advanced approach that expands the scope of molecular docking studies, allowing for a comprehensive exploration of protein and protein–ligand complex conformational stability over specified timeframes, typically measured in nanoseconds. By employing MD simulations, researchers can validate and support the findings obtained through molecular docking experiments. Several crucial parameters are employed to thoroughly assess the quality of molecular docking analysis. One of these key parameters is the RMSD, which quantifies the average displacement between the experimentally determined structure and the simulated conformation during MD simulations. Another significant metric is the Rg, which assesses the compactness and overall size of the biomolecular system throughout the simulation. Additionally, the RMSF reveals the flexibility and dynamic behavior of individual atoms or residues within the system over time. This parameter aids in the identification of regions that exhibit significant fluctuations or structural changes during the simulation.

Furthermore, protein–ligand contacts plots play a vital role in evaluating the stability and strength of various interactions within the protein–ligand complex during MD simulations. This analysis provides valuable insights into the nature of key intermolecular interactions that are crucial for ligand binding and protein function. Molecular dynamics serves as a valuable tool for gaining a deeper understanding of the conformational dynamics and stability of biomolecular systems. The utilization of parameters such as RMSD, Rg, RMSF, and protein–ligand contacts plots is essential for rigorously assessing the reliability and accuracy of molecular docking investigations throughout the 100 ns MD trajectory.

#### 3.2.1. RMSD Plot Scrutiny

The RMSD plot visually depicts the average displacement of a specific set of atoms within a target molecule relative to a reference frame. It offers a graphical representation of how the protein’s structure changes over the course of the simulation. The trajectory of these structural dynamics is observed from 0 ns to 100 ns.

GSK-3β RMSD was noted from 2.5 Å at the initial time point, and an upsurge of 1.5 Å in the graph was noticed at 22 ns. Again, an exponential decrease in the spike was observed from 23 to 38 ns of approximately 2.3 Å, a further increase and decrease in spikes was observed until 80 ns, and a platonic phase was noticed from 81 to 95 ns at 2.9 Å and was finally equilibrated to 3.5 Å at 100 ns, as depicted in Figure 8a. Though there is an increase and decrease in the graph during the end of the time point (100 ns), the graph is observed to be stable. This indicates that the protein–ligand interactions are quite stable and can be considered for further study. Moreover, in the RMSD of BuChE, initially the value was observed from 1.75 Å and a spike was observed to 2.25 Å at 20 ns. A constant decrease and increase in the spikes were noted from 21 to 80 ns at 2 Å and finally equilibrated at 2 Å until the end of 100 ns MD duration, as depicted in Figure 8b. The protein–ligand complexes (GSK-3β-DSK1 and BuChE-DSK4) revealed a conformational stability of ligands in the active site of the protein based on the plot analysis; this analysis thus indicates a good protein–ligand interaction.

#### 3.2.2. Protein RMSF Plot Analysis

RMSF is another parameter that reveals the quality of molecular dynamic simulation indicating the amino acid residues average fluctuations over the time scale from the initial coordinates. RMSF is the average position or fluctuation of amino acids from the initial coordinates over a time scale. Figure 9a,b depict the RMSF of the amino acids of GSK-3β and BuChE, in which a α-helical structure is indicated in magenta, while β-sheets are in green The red color is coded for the amino acids present in the active site, which have the potential chance of interaction with the inhibitor. Higher fluctuations have been observed for terminal amino acids, and they were not in the vicinity of the binding site. However, minor insignificant conformational changes could be noticed in the active site and chain. Overall, the fluxes of amino acid residues that have proximity with inhibitors were within the tolerable range, which is less than 2.0 Å.

#### 3.2.3. Protein RMSF Plot Analysis

In the protein RMSF plot, minimal fluctuations were noticed for DKS1 and for DKS4, and it revealed some higher fluctuations for atom number 8 (aromatic carbon of pyridopyrimidine) and 41–43 (-F atoms of CF_3_) in the overall of dynamics study, as shown in Figure 10a,b.

#### 3.2.4. Protein–Ligand Interactions

One vital aspect in assessing the solidity of the protein–ligand complex is the analysis of the interactions of amid ligands and amino acids within the protein’s active site. These interactions engross various types, including hydrophobic, electrostatic, hydrogen bond, ionic, and water bridge polar interactions. Scrutiny of protein–ligand interactions showed that ionic, hydrophobic interactions contributed with an optimal percentile and the water bridging interactions contributed with a higher percentile. In the due course of the MD analysis, the DKS1 with GSK-3β demonstrated water bridging polar contacts with Ser66 (44%) and Glu137 (33%); four hydrogen bond interactions were observed with Val135 (66, 89, and 99%) and Cys199 (72%) and hydrophobic π-π staking contacts with Phe67, as stated in Figure 11a,b. DSK4 with BuChE revealed five polar hydrogen bond contacts with Gly116 (74%), 117 (72%), Asp70 (58%), Glu197 (86%), and Ala328 (50%); hydrophobic and electrostatic contacts with Trp82 (82%, π-π staking) and Phe329 (60%, π-π staking); and π-cation interaction with His438 (95%). The maximum number of water bridging interaction were spotted with Pro285 (43; 90%), Asn68 (62%), and Trp430 (30%), as portrayed in Figure 11c,d.

For the conclusion of the MD studies, the protein structures of GSK-3β and BuChE were analyzed for their structural stability. As depicted in Figure 12a,b, a quint alteration in the crystal structure of GSK-3β was noticed in the C-terminal arena, and in the Cα-helix no alternations in the active site topology of the macromolecule were spotted in the amid timeline of 0 to 100 ns. Similarly, for the macromolecule BuChE, a conformational change was visible in the envelope region above the active pocket of the protein, with no fluctuations in the binding site of BuChE from 0 to 100 ns of the MD scrutiny, as shown in Figure 13a,b. Overall, it was clear that both the protein X-ray crystal structures portrayed minimal shifts in their conformational solidity through the due course of the MD analysis.

### 3.3. Molecular Dynamic Cross-Correlation and PCA Scrutiny

The “principal component” of the PCA provides the protein–ligand simulation dynamics. From the principal component, one can identify the structural and dynamical properties of the protein–ligand complex. It gives an idea about the binding pocket, ligand induced conformation change, and more. Figure 14a,b depict the heat maps of the compound DKS1 and DKS4, respectively. Figure 15a,b represent the 2D line graphs of DKS1 and DSK4 from these graphs, showing the stability of the protein–ligand complex. From the PCA analysis, we obtained a set of data, which are reduced to this graph by taking two values across X and Y, respectively. Interpreting the two graphs, we can see that the values attain stability at the end of the analysis. This observation gives us a valuable insight into the conformation of the protein–ligand, in that there is not much change in the conformation.

The graph plotted using the PCA1 and PCA2 can be seen as having a negative correlation for DKS1 in the initial time duration, but it can be seen to have correlated positively for the majority of the time duration. For the plot of DKS4, a negative result is seen initially, which can be observed to have undergone a sudden increase and decrease in the values. However, a positive correlation can be seen at the end of the simulation time. Therefore, clearly, we can see that the frames become steady at the end of the simulation time.

### 3.4. In Silico ADME Profiling

Schrödinger’s QikProp module has proven to be a valuable tool for accurately predicting the ADMET properties of designed molecules. In Table 2, the evaluated molecules are listed along with their corresponding properties. Lipinski’s rule of five was employed, which considers factors like molecular weight, polar surface area, hydrogen bond donor and acceptor counts, and calculated octanol water partition coefficients. All the compounds studied successfully met the criteria outlined in Lipinski’s rule of five, namely having fewer than 5 hydrogen bond donors, fewer than 10 hydrogen bond acceptors, and a QPlogP value of less than 5. All the molecules from the series demonstrated molecular weight arraying from 494.53 to 627.987 daltons, which does not fall within the limits of the Lipinski rule of five. All the molecules portrayed a maximum count of 5 HBA and 10 HBD, which also was in the acceptable limit of the Lipinski rule of five. The TPSA of the molecules under scrutiny demonstrated values in the range of 140.804 to 177.621 Å2 and were within the prescribed range of the Lipinski rule of five. Based on the data, all the molecules did not violate any one of the Lipinski rules of five. All the newly designed molecules of the series illustrated similar oral absorption scores in the range of 29.409 to 79.792%, from which compound DKS5 revealed a higher percentage of oral absorption of 79.792. In conclusion, all the compounds qualified for the ADMET parameters, with an acceptable standard error prescribed, as stated in Table 2.

## 4. Conclusions

Alzheimer’s disease is a serious and widespread neurodegenerative condition. Unfortunately, the current drugs available for managing Alzheimer’s have shown limited effectiveness, leading to a push for innovative therapeutic approaches, particularly the development of novel dual inhibitors. The primary goal of this study was to design drug-like molecules that not only exhibit improved efficacy but also minimize the potential side effects. Researchers have conducted extensive molecular docking investigations involving both GSK-3β and BuChE. This analysis revealed that the compounds DKS1 and DKS4 exhibited remarkable molecular interactions with the active sites of these enzymes. These interactions were found to be crucial, involving essential amino acids and resulting in favorable docking energies of −9.6 and −12.3 kcal/mol for DKS1 and DKS4, respectively. To further assess the stability of DKS1 and DKS4 within the active cavities of GSK-3β and BuChE, researchers have also performed extensive molecular dynamics simulations. These simulations have confirmed the sustained binding of these ligands, suggesting their potential as promising candidates for therapeutic intervention. While evaluating the pharmacological profile of these compounds, it was found that the molecules demonstrating the most favorable docking behavior also possessed excellent ADMET properties. Notably, DKS5 displayed an exceptional human oral absorption rate of 79.792%, outperforming the other molecules in this regard. Overall, the in silico studies conducted in this research unveiled the potential of the designed molecules to effectively inhibit both GSK-3β and BuChE. These findings offer a promising strategy for the development of novel lead compounds aimed at mitigating the devastating impact of AD.

## Figures and Tables

**Figure 1 pharmaceutics-16-00991-f001:**
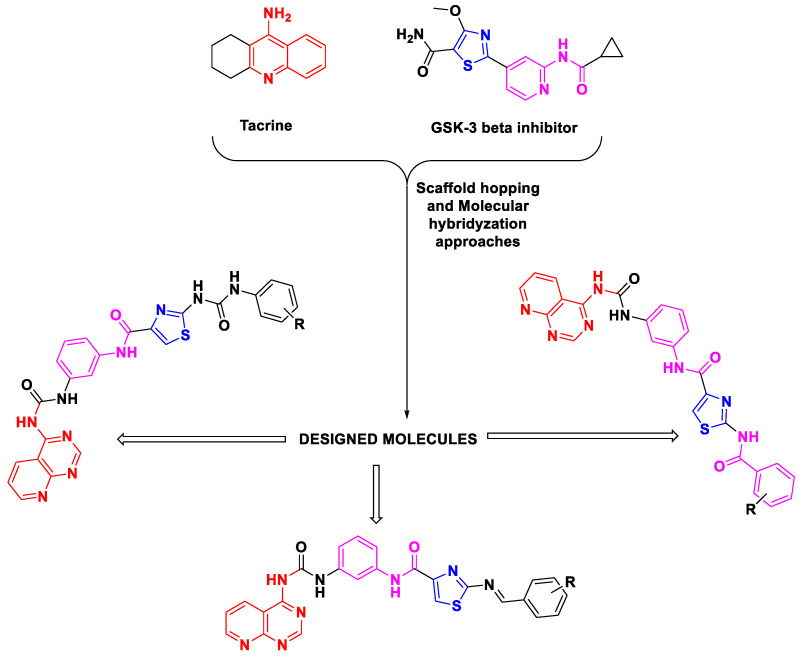
Designing strategy of dual inhibitors DKS the chained GSK-3β and BuChE pharmacophores Tacrine and thiazole motif by the application of molecular hybridization and scaffold hopping methodology.

**Figure 2 pharmaceutics-16-00991-f002:**
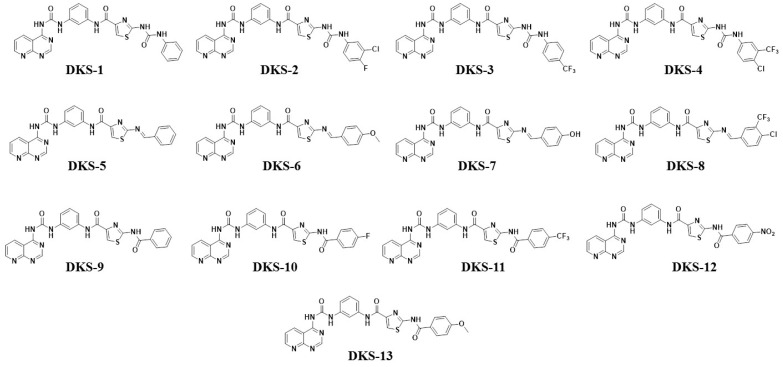
Chemical structures of designed molecules.

**Figure 3 pharmaceutics-16-00991-f003:**
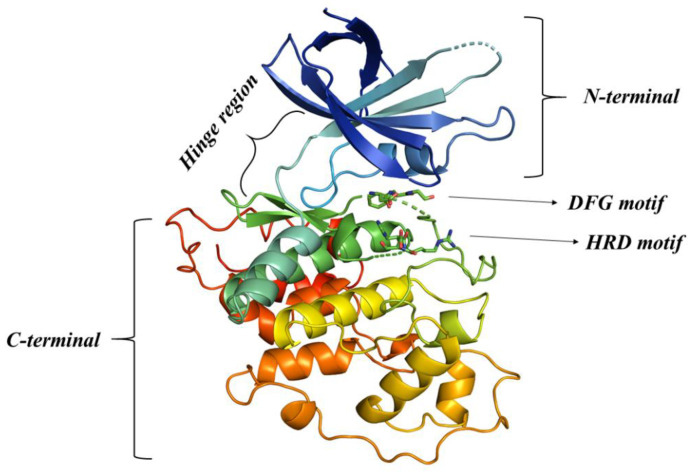
Molecular architecture of GSK-3β.

**Figure 4 pharmaceutics-16-00991-f004:**
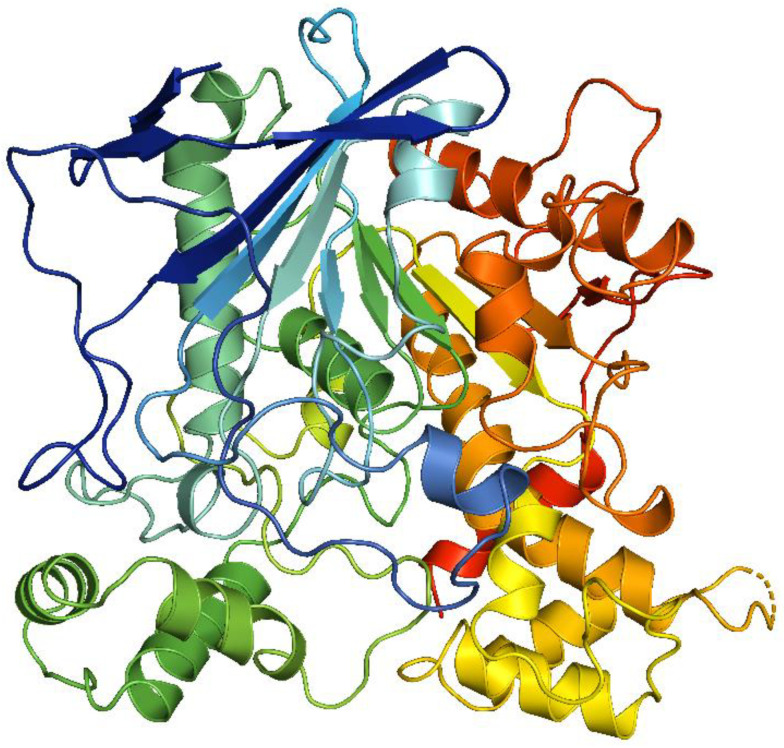
Molecular skeleton of BuChE.

**Figure 5 pharmaceutics-16-00991-f005:**
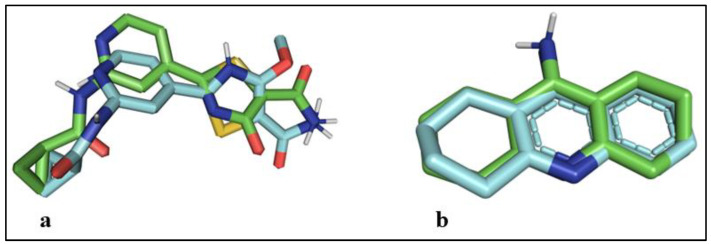
Coinciding poses of the co-crystallized ligands: (**a**) docked conformation of 2WE in cyan with co-crystallized ligand in green, RMSD: 0.98 Å; (**b**) docked conformation of THA (Tacrine) in cyan with co-crystallized ligand in green, RMSD: 0.5 Å.

**Figure 6 pharmaceutics-16-00991-f006:**
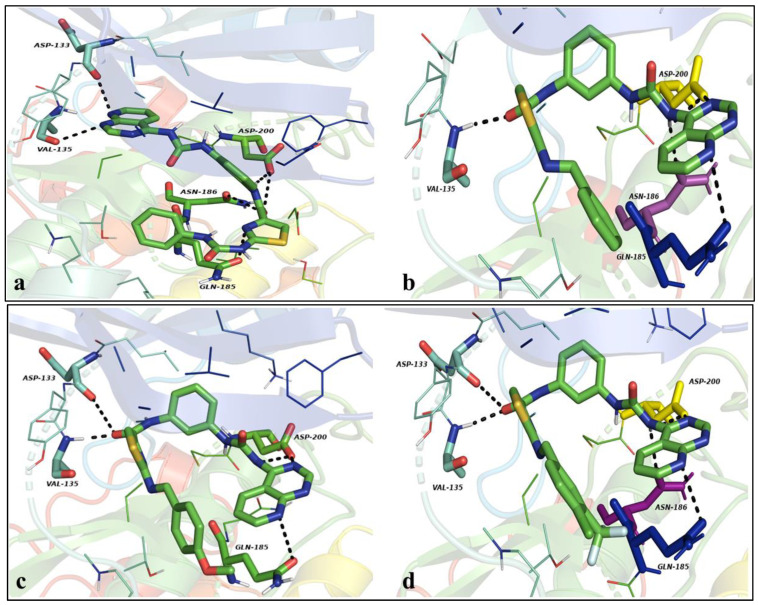
(**a**–**d**) Docked complexes of DKS-1, -5, -6, and -8 with GSK-3β kinase.

**Figure 7 pharmaceutics-16-00991-f007:**
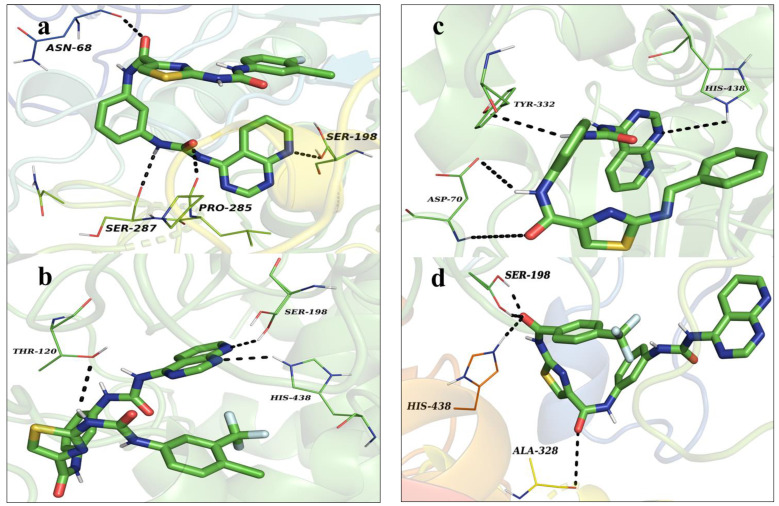
(**a**,**b**) Docked complexes of DKS-2 and -4 with BuChE. (**c**,**d**) Docked complexes of DKS-5 and -11 with BuChE.

**Figure 8 pharmaceutics-16-00991-f008:**
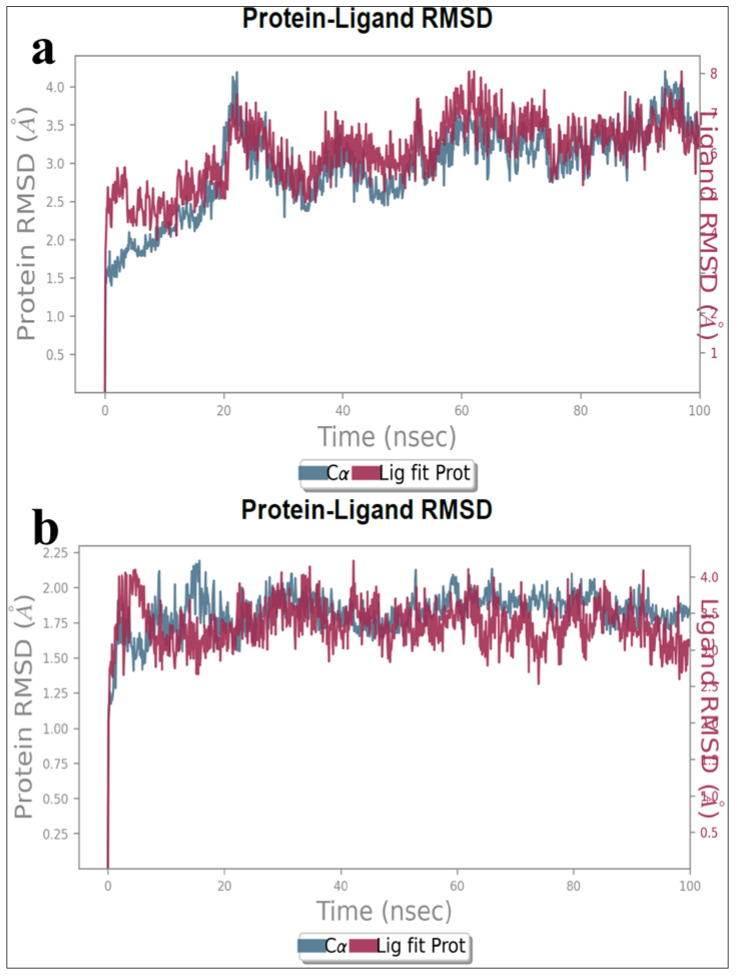
(**a**,**b**) RMSD plot of DSK1 complexed with GSK-3β and DSK4 complexed BuChE.

**Figure 9 pharmaceutics-16-00991-f009:**
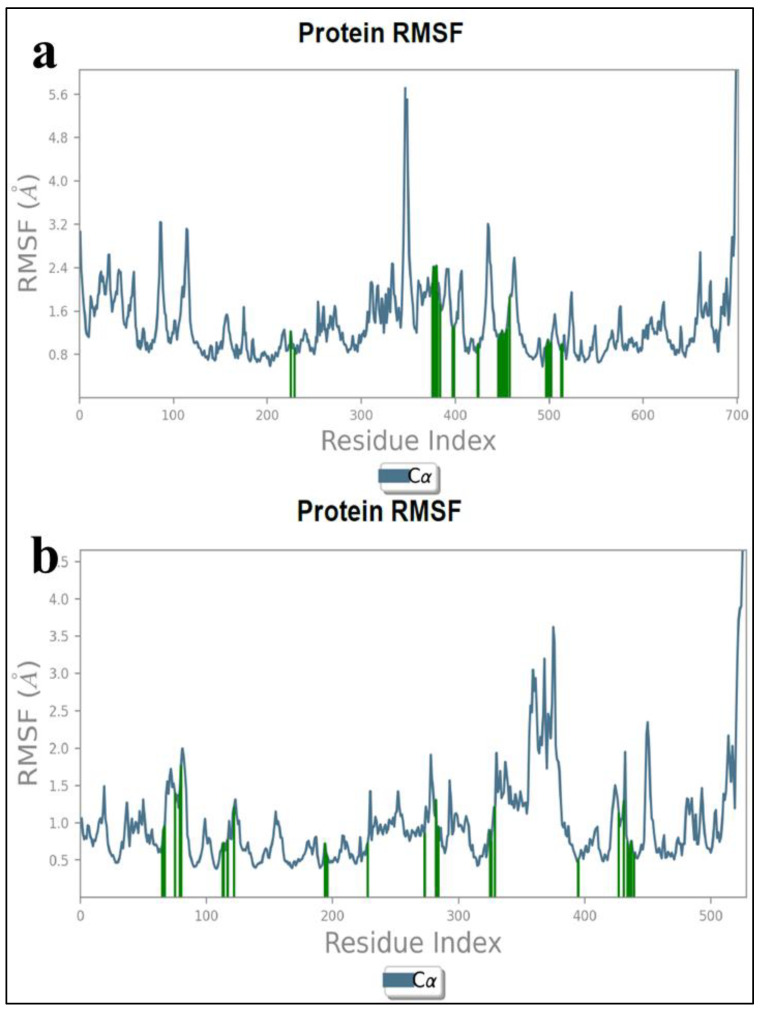
(**a**,**b**) RMSF plot of DSK1 complexed with GSK-3β and DSK4 complexed BuChE.

**Figure 10 pharmaceutics-16-00991-f010:**
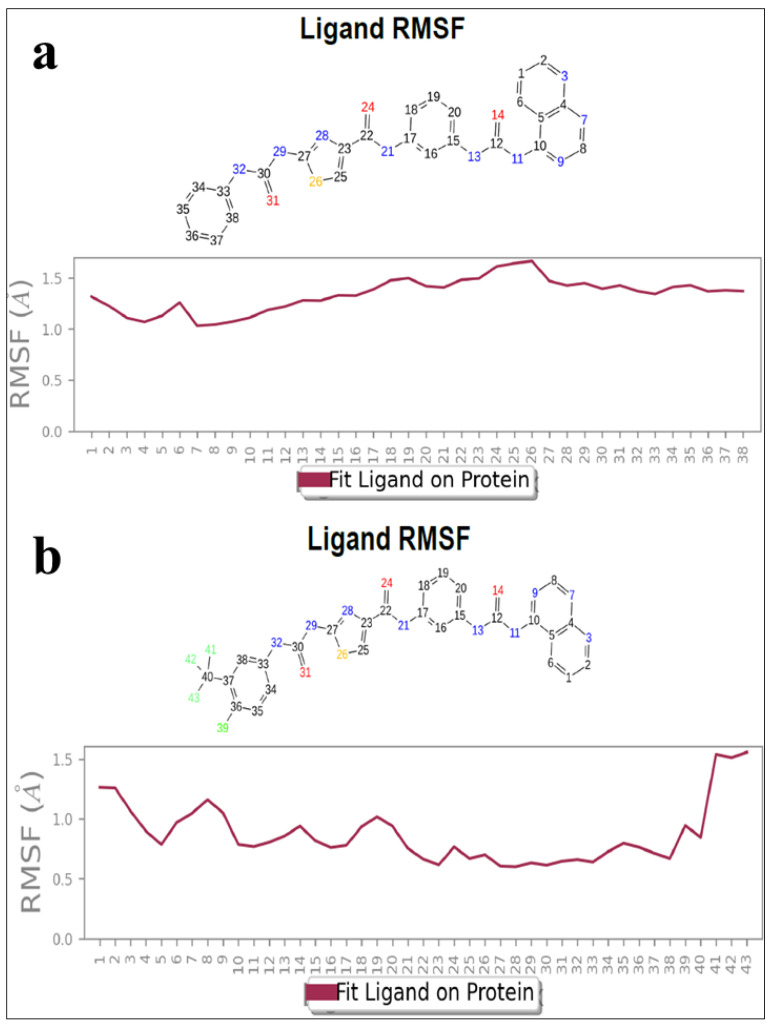
(**a**,**b**) RMSF plot of DSK1 complexed with GSK-3β and DSK4 complexed BuChE.

**Figure 11 pharmaceutics-16-00991-f011:**
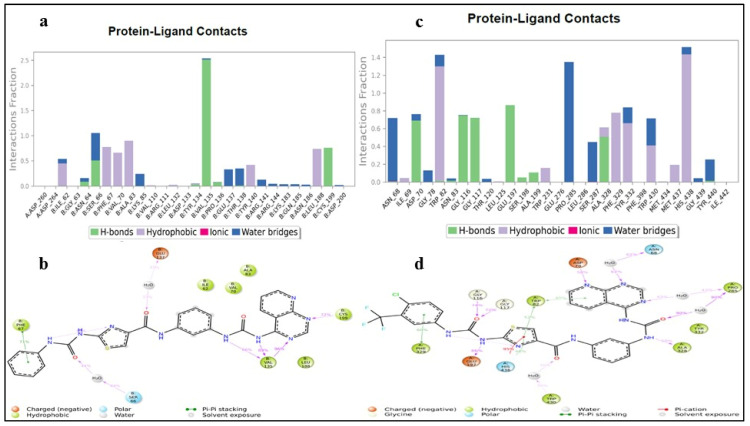
(**a**–**d**) Protein–ligand contact plot of DSK1 complexed with GSK-3β and DSK4 complexed with BuChE.

**Figure 12 pharmaceutics-16-00991-f012:**
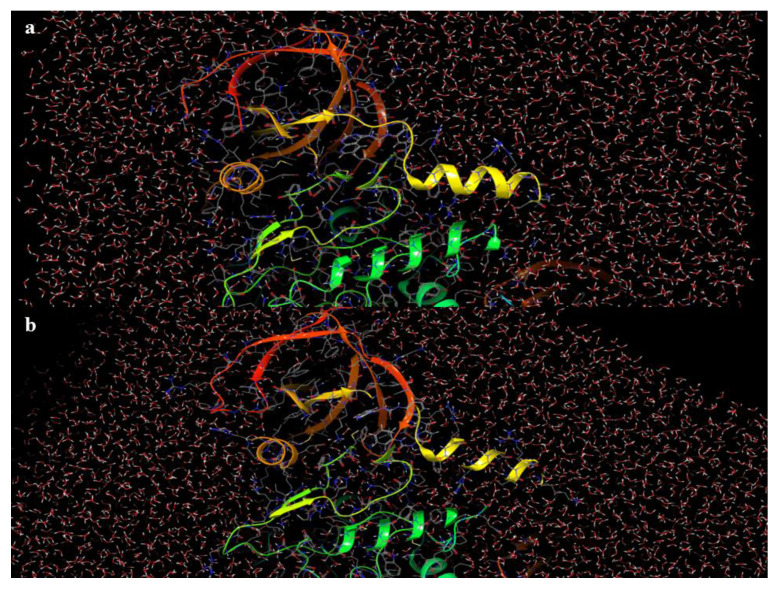
(**a**,**b**) Crystal structure of BuChE at 0 and 100 ns.

**Figure 13 pharmaceutics-16-00991-f013:**
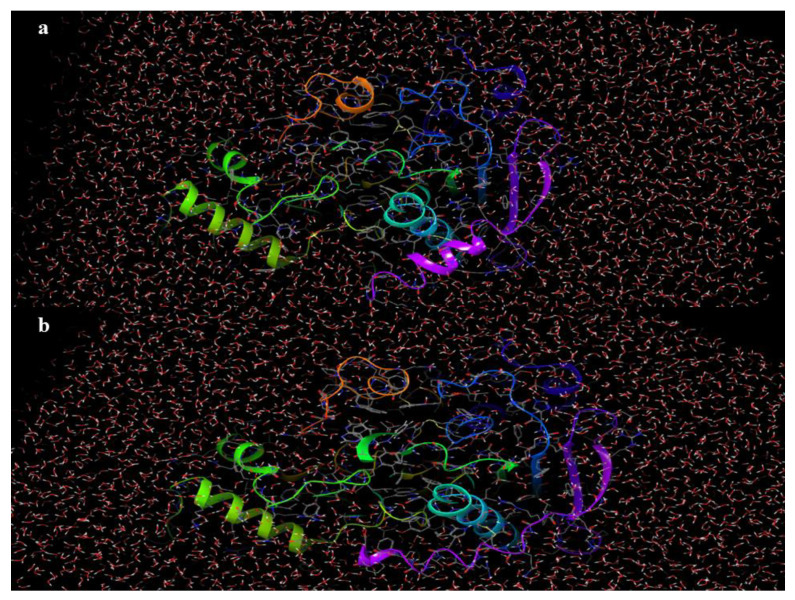
(**a**,**b**) Crystal structure of GSK-3β at 0 and 100 ns.

**Figure 14 pharmaceutics-16-00991-f014:**
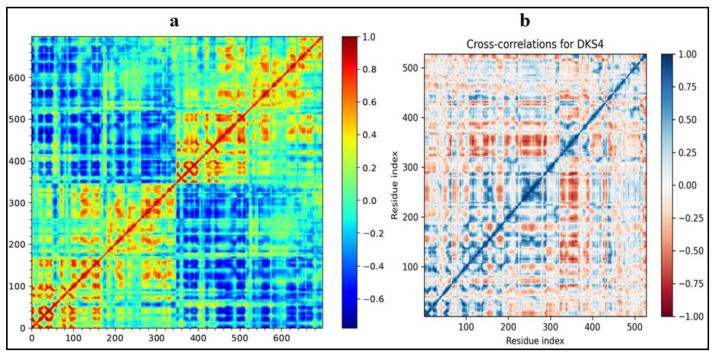
(**a**,**b**) Heat maps of DKS1 and DKS4.

**Figure 15 pharmaceutics-16-00991-f015:**
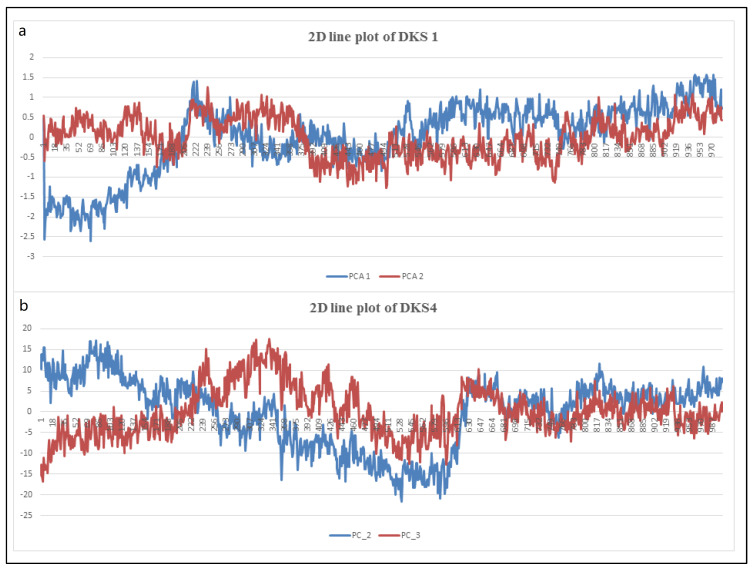
(**a**,**b**) 2D graphical notation of DKS1 and DKS4.

**Table 1 pharmaceutics-16-00991-t001:** Molecular interactions and docking energies of the molecules.

Sr No.	Molecule Id	Hydrogen Bond Interactions	Hydrophobic Interactions	Docking Energy (kcal.mol^−1^)
GSK-3β kinase (PDB ID: 4PTC)				
1	DKS-1	Val135, Asp133 and Asp200	Arg141, Thr138, Tyr140, Gln185, Lys183, Asn186, Gly65, Asn64, Phe67, Gly63, Val70, Ala83, Ile62 and Asp200	−9.6
2	DKS-5	Val135, Asp200 and Asn186	Arg141, Ile62, Tyr134, Asp133, Phe67, Lys85, Val70, Cys199, Leu132, Gln185, Ala83, Leu188, Thr138 and Tyr140	−9.1
3	DKS-6	Val135, Asn186 and Asp200	Arg141, Ile62, Leu188, Tyr134, Ala83, Asp133, Leu132, Val70, Cys199, Lys85, Phe67, Val110, Gln185, Thr138 and Tyr140	−8.6
4	DKS-8	Val135, Asn186, Asp200, Thr138 and Arg141	Tyr134, Leu188, Ile62, Asp133, Phe67, Val70, Lys85, Cys199, Leu132, Gln185, Ala83, Val110 and Tyr140	−9.3
BuChE (PDB ID: 4BDS)				
5	DKS-2	Tyr332, His438 and Ser198	Leu125, Thr120, Gln119, Pro285, Gly115, Gly117, Trp430, Met437, Tyr440, Ala328, Gly439, Asp70, Ser287, Val288, Leu286, Phe398, Glu197 and Tyr128	−11.4
6	DKS-4	His438, Ser198 and Thr120	Phe329, Phe398, Ser287, Val288, Gly116, Ala328, Tyr332, Asp70, Gln119, Asn68, Pro285, Thr284, Trp82, Tyr440, Gly439, Glu197, Gly117, Trp231 and Leu286	−12.3
7	DKS-5	Tyr332, Asp70 and His438	Phe398, Trp231, Gly116, Ser198, Phe329, Gly439, Glu197, Trp82, Tyr440, Ala328, Trp430, Ser79, Tyr332, Pro285, Thr120, Gly117, Leu286, Val288 and Met437	−12.1
8	DKS-11	Ser198, His438 and Asn289	Gly116, Ser287, Gly117, Pro285, Val288, Leu286, Phe329, Asp70, Thr120, Glu276, Ala277, Asn68, Gln119, Ala328, Tyr332, Trp82 and Glu197	−11.7

**Table 2 pharmaceutics-16-00991-t002:** ADMET properties of designed molecules.

Lipinski Rule of 5							Jorgensen’s Rule of Three				In Silico Predicted ADMET				
Compd ID	Mol. MW	Donor HB	Accpt. HB	QPlogP o/w	PSA	Violation	QP logS	QPP Caco	#metab	Violation	QP logBB	QPP MDCK	QP logKhsa	SASA	% Human Oral Absorption
DKS1	525.544	5	11.5	1.404	177.621	2	−5.087	13.38	7	2	−1.674	207.462	−0.507	771.864	29.409
DKS2	577.979	5	11	2.067	177.533	2	−6.051	13.713	6	2	−2.581	104.647	−0.383	891.521	33.482
DKS4	627.987	5	11	2.725	177.482	2	−6.921	13.585	7	3	−2.511	213.677	−0.204	930.794	37.264
DKS5	494.53	3	10	2.95	141.173	0	−6.21	97.162	6	1	−2.095	109.926	−0.063	843.677	79.792
DKS6	524.556	3	10.75	3.061	149.355	2	−6.457	97.237	7	2	−2.206	111.617	−0.056	883.728	54.532
DKS8	596.973	3	10	4.317	140.804	1	−8.135	99.448	7	2	−1.776	930.772	0.269	915.699	75.019
DKS11	578.527	4	11	2.352	164.778	2	−5.214	51.343	6	0	−2.795	25.86	−0.182	860.721	45.414

## Data Availability

Data is contained within the article.

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
