# Peer review of "Computational and ADMET Predictions of Novel Compounds as Dual Inhibitors of BuChE and GSK-3β to Combat Alzheimer’s Disease"

_pharmaceutics, 2024, doi:10.3390/pharmaceutics16080991_

Round 1

Reviewer 1 Report

Comments and Suggestions for Authors

Computational and ADMET predictions of novel compounds as dual inhibitors of BuChE and GSK-3β to combat Alzheimer's disease is found to be very interesting. However, authors have to address important experimental finding. This article is recommended for major revision.

Authors must address following comments before submitting revised manuscript.

1. Authors have included few compounds ie. DKS-1,DKS-4 so on. Please provide the details of these compounds both experimental and spectral data.

2. Why all the compounds used for molecular docking.

4. What is the standard used for the molecular docking?

5.  Include exact structures of those compounds.

6. Figure 1: Provide standard and recent reference for in silico studies.

7. Images of docked complexes can be shown in better way.

8.  include references for Protein RMSF plot analysis and swiss ADME.

9. Conclusion part be better written.

10. Whether these compounds are toxic in nature?

11. Also, report those are bioavailable or not.

Comments on the Quality of English Language

 Moderate editing of English language required

Reviewer 2 Report

Comments and Suggestions for Authors

1.       Authors affiliations are not in sequence. 3rd author got affiliation number 3 but suppose to get 2. Do the corrections as suggested.

2.       In abstract, the language is overly complex and verbose, making it difficult to read and understand. The background section is vague, lacking specific information about the significance of targeting GSK-3β and BuChE in Alzheimer's disease. The methodology is too brief and lacks essential details, such as key steps and parameters. The results section is unclear and does not provide context or significance for the findings, lacking a comparison with existing treatments. The molecular dynamics simulations are mentioned without sufficient context or explanation of their significance. The conclusion is overly optimistic without providing concrete evidence or next steps. Additionally, there is inconsistent focus between different compounds, leading to potential confusion. The abstract also contains grammatical errors and stylistic issues, such as inconsistent tense usage and unnecessary repetition.

3.       It does not provide sufficient context for why GSK-3β and BuChE are significant targets for Alzheimer's disease, and it should briefly explain their roles and relevance to the disease more clearly. The specific objectives and hypotheses of the study are not clearly stated, and the rationale behind using hybrid ligands and the specific choice of tacrine and adamantane is not clearly justified. The text repeats information unnecessarily, such as the significance of GSK-3β in tau phosphorylation and the role of BuChE in acetylcholine degradation, which can be minimized to maintain conciseness. While some methodological aspects are mentioned, they are not presented in a coherent manner, and the introduction should briefly outline the main methodological approaches without going into excessive detail. Additionally, there are several grammatical errors and stylistic issues, including inconsistent use of tense and awkward phrasing, which detract from the overall readability and professionalism of the text. Lastly, the introduction references previous studies but does not sufficiently integrate these references into the narrative to build a strong case for the current research.

4.       There is redundancy and unnecessary repetition of information. For instance, the description of the molecular docking tool AutoDock Vina and the validation process are repeated, which could be streamlined to improve readability.

5.       on ADMET profiling lists numerous parameters but does not provide a clear rationale for why these specific parameters were chosen or how they were calculated. There is also no mention of the criteria for selecting the 14 ligands used in the study, nor is there an explanation of how these ligands were prepared.

6.       The details provided for the molecular interactions are not comprehensive. For instance, while the docking energy scores are mentioned, the specific conditions under which these scores were obtained are not clearly explained. There is also a lack of explanation about the significance of these interactions in the context of the study's objectives.

7.       The section is heavy on technical jargon without sufficient explanation. Terms like "π-sigma interactions," "π-π stacking contacts," and "π-alkyl contacts" are used without clarifying their importance or how they influence the docking results.

8.       The results for GSK-3β and BuChE are presented in a disjointed manner. For example, the results for molecule DKS-1 are followed by a brief mention of DKS-5 and DKS-6, without a clear transition or detailed comparison. This makes it hard to understand the relative performance of different molecules.

9.       The validation of the docking protocol is mentioned, but the details are sparse. The re-docking experiment is described briefly, but there is no detailed discussion about the validation metrics or how they confirm the reliability of the docking results.

10.   The RMSD plot analysis for GSK-3β and BuChE is poorly described. The narrative is confusing and the description of the RMSD changes over time lacks clarity. The text jumps between different time points without a clear explanation of the trends observed and their implications.

Comments on the Quality of English Language

English correction is required

Reviewer 3 Report

Comments and Suggestions for Authors

This provides computational approaches including docking studies and ADMET predictions of novel compounds as dual inhibitors of BuChE and GSK-3b. It may give important insights for Alzheimer's disease. The presented scientific facts and discussions might be valuable of being published in Pharmaceutics. Rather than scientific contents, ways of presentation need revisions. Please see below.

1) ADMET has to be explained somewhere (maybe in Introduction?) in the early part of this manuscript.

2) Use of too many abbreviations may make it difficult to read this manuscript. In may be a good idea to add summary table for abbreviations.

3) Reaction schemes in Figure 1 are somehow questionable, because these are not formal reaction equation (leaving groups are not described). Please supply more convincing reaction schemes.

4) In figures for docking studies, representations of amino acid residues are too small. Please use larger font size.

5) Quality of Figures 7 to 10, and 14 are not so high (some words are too small).

Please improve quality of these figures.

Round 2

Reviewer 1 Report

Comments and Suggestions for Authors

Manuscript is recommended for publication 

Comments on the Quality of English Language

Moderate editing is required